# Profiling surface proteins on individual exosomes using a proximity barcoding assay

Di Wu [1,2,3], Junhong Yan[1], Xia Shen [4,5,6], Yu Sun [7,11], Måns Thulin [8,12], Yanling Cai[9], Lotta Wik[1], Qiujin Shen[1], Johan Oelrich [3], Xiaoyan Qian [2], K. Louise Dubois[10], K. Göran Ronquist [10], Mats Nilsson[2], Ulf Landegren [1] & Masood Kamali-Moghaddam [1]

Exosomes have been implicated in numerous biological processes, and they may serve as important disease markers. Surface proteins on exosomes carry information about their tissues of origin. Because of the heterogeneity of exosomes it is desirable to investigate them individually, but this has so far remained impractical. Here, we demonstrate a proximity-dependent barcoding assay to profile surface proteins of individual exosomes using antibody-DNA conjugates and next-generation sequencing. We first validate the method using artificial streptavidin-oligonucleotide complexes, followed by analysis of the variable composition of surface proteins on individual exosomes, derived from human body fluids or cell culture media. Exosomes from different sources are characterized by the presence of specific combinations of surface proteins and their abundance, allowing exosomes to be separately quantified in mixed samples to serve as markers for tissue-specific engagement in disease.

[1] Department of Immunology, Genetics and Pathology, Science for Life Laboratory, Uppsala University, SE-75 185 Uppsala, Sweden. [2] Department of Biochemistry and Biophysics, Science for Life Laboratory, Stockholm University, SE-17 165 Solna, Sweden. [3] Vesicode AB, Nobels väg 16, Solna SE-17 165, Sweden. [4] Center for Global Health Research, Usher Institute of Population Health Sciences and Informatics, University of Edinburgh, Teviot Place, Edinburgh EH8 9AG, UK. [5] Department of Medical Epidemiology and Biostatistics, Karolinska Institutet, Nobels väg 12 A, Stockholm SE-17 177, Sweden. [6] Biostatistics Group, State Key Laboratory of Biocontrol, School of Life Sciences, Sun Yat-sen University, Guangzhou CN 510000, China. [7] Department of Molecular Evolution, Uppsala University, SE-751 24 Uppsala, Sweden. [8] Department of Statistics, Uppsala University, SE- 75 120 Uppsala, Sweden. [9] Institute of Translational Medicine, The Second People's Hospital of Shenzhen, Shenzhen CN 518000, China. [10] Department of Medical Sciences, Clinical Chemistry, Uppsala University, SE-75 185 Uppsala, Sweden. [11] Present address: Guangdong Provincial Key Laboratory of Protein Function and Regulation in Agricultural Organisms, College of Life Sciences, South China Agricultural University, Guangzhou, Guangdong 510642, China. [12] Present address: School of Mathematics and Maxwell Institute for Mathematical Sciences, University of Edinburgh, Teviot Place, Edinburgh EH8 9AG, UK. Correspondence and requests for materials should be addressed to D.W. (email: di.wu@vesicode.com)

Exosomes are a subclass of membrane-coated extracellular vesicles with sizes of 30–100 nm, which are released from cells by exocytosis. Exosomes are found in most body fluids, and they have been shown to play key roles in processes such as coagulation, intercellular signaling, immune responses, and cellular waste management[1]. Compelling evidence suggests that exosomes may have a role in the spread of cancer from a primary tumor to metastasis sites[2,3], and they are promising as tissue-specific biomarkers for liquid biopsy[4,5]. Exosomes are highly heterogeneous in molecular composition[6–9], and their surface proteins bear characteristics of their tissues of origin[10,11], rendering specific subclasses of these vesicles promising to demonstrate pathology affecting specific tissues[12–14]. Accordingly, it is important to investigate exosomes individually as such information may be lost in bulk-level analyses.

Recently, new technologies have emerged that improve opportunities for detection of exosomes. Imaging flow cytometry overcomes obstacles in traditional flow cytometry by including a CCD camera with a 60× objective, allowing detection of vesicles with sizes below 500 nm through enhanced fluorescence[15]. However, only a small number of fluorophore-labeled antibodies can be resolved in this way. Nano-plasmonic sensors utilize sophisticated nanohole arrays to first isolate single exosomes via specific capture antibodies, followed by protein profiling using detection antibodies[16]. The combination of capture and detection antibodies limits the analysis to two protein targets per exosome, and sandwich immune assays are limited to analyzing pairs of proteins on exosomes in bulk. Therefore, methods are needed to more comprehensively profile proteins in high multiplex for individual exosomes.

DNA-assisted immunoassays combine affinity probes with conjugated amplifiable oligonucleotides, converting protein identities to DNA sequences for protein detection even at the level of single molecules or molecular complexes. In particular, proximity ligation or extension assays can offer improved specificity of analysis and confer information about protein compositions via ligation or extension of pairs of DNA strands brought in proximity via their conjugated antibodies[17–19]. A multiple-recognition proximity ligation assay (4PLA) has been developed where exosomes are captured by an immobilized antibody, whereupon four antibody-DNA conjugates give rise to amplifiable DNA strands for highly specific and sensitive detection of prostate-derived exosomes—prostasomes[12]. Similarly, antibody-DNA conjugates have been used for flow-cytometric detection of individual exosomes by pairwise ligation of several sets of antibody-DNA conjugates enhanced via rolling circle amplification (RCA) for fluorescence detection[20]. However, new technologies are required to survey higher orders of protein species on large sets of exosomes in parallel in order to assess their heterogeneity.

Here, we report a proximity-dependent barcoding assay (PBA), as a high-throughput approach to simultaneously profile 38 surface proteins for their presence on individual exosomes. In PBA, we use micrometer-sized single-stranded DNA clusters, each having hundreds of copies of a unique DNA motif, generated via RCA, to barcode individual exosomes. The protein composition on the surface of individual exosome is converted to DNA sequence information via bound antibody-DNA conjugates that incorporate a random tag sequence repeated in each RCA product. After amplification by PCR, information about protein and exosome identity brought together in DNA strands is decoded by next-generation sequencing to identify the surface protein composition of individual exosomes.

## Results

### Design and workflow of PBA
PBA probes were prepared by conjugating antibodies with DNA oligonucleotides comprising a 8-nucleotide (nt) proteinTag that served to identify the target exosomal surface protein, and a 8-nt random unique molecular identifier (UMI) sequence, here referred to as a molecule tag (moleculeTag), to distinguish individual protein molecules after PCR amplification[21] (Fig. 1a). Reagents to barcode individual exosomes were prepared by RCA of circularized DNA molecules containing a 15-nt random DNA sequence (complexTag). Each RCA product includes several hundred identical copies of a unique complexTag (Fig. 1b), and in the PBA procedure these become incorporated in the antibody-conjugated oligonucleotides, and serve to identify proteins on individual exosomes having become colocalized with unique RCA products as described in Fig. 1[22].

In PBA, exosomes are mixed with PBA probes in solution before being captured sparsely in 96-well microtiter wells coated with cholera toxin subunit B (CTB) that binds GM1 gangliosides in the exosome membranes[23], followed by washes to remove free antibody conjugates (Supplementary Fig. 1). Diluted RCA products are then added, allowing individual RCA products, similar in size to exosomes (Supplementary Fig. 2) to interact with single exosomes by having their bound PBA probes hybridize to the RCA products. PBA probes bound to the same exosome are brought in proximity, and incorporate the same complexTag from a nearby RCA product by DNA polymerase-mediated extension (Fig. 1c). Next, the extension products are amplified for preparation of a sequencing library. Neither oligonucleotides on exosomes that have failed to encounter an RCA product nor isolated RCA products can give rise to amplifiable products. After sequencing amplification products, the reads of each sample are sorted by complexTags and proteinTags to identify participating proteins on individual exosomes. By counting the total number of different moleculeTags for each of the proteinTags sharing the same complexTag, all detected protein molecules from individual exosomes in a sample can be identified and quantified.

### Validation of PBA using a separate or mixed incubation system
To ascertain that PBA could correctly identify members of protein complexes, we first prepared artificial complexes by allowing streptavidin (STV) to bind combinations of biotinylated oligonucleotides (Fig. 2a). We incubated STV with oligonucleotides carrying each one of four proteinTags in four separate preparations, such that each STV in the mixture bound oligonucleotides with the same proteinTag before these four STV preparations were combined (separate incubation). If two different STV-oligonucleotide complexes would be erroneously identified by PBA as a single one, then there is an ~75% possibility that such falsely identified complexes would involve more than one proteinTag. For comparison, we also incubated STV with a mixture of all four biotinylated oligonucleotides, such that each STV would bind some combination of these four oligonucleotides (mixed incubation). STV were incubated with biotinylated oligonucleotides at either a 1:10 or 10:1 molar ratios. In the analysis of samples from separate preparations, less than 1% of the recorded complexes included more than one proteinTag. In the mixed incubation samples with oligonucleotides in excess, 21.0% of the complexes were shown to include two distinct proteinTags, while 2.4% and 0.2% of complexes had three or four proteinTags, respectively. As expected, we observed more complexes with more than one oligonucleotide per STV in preparations where oligonucleotides were in excess of STV. In these preparations, the recorded complexes containing 2, 3, or 4 moleculeTags from either separate or mixed preparations were sorted according to the proteinTags per complex (Fig. 2b). The majority of the separately prepared complexes contained only 1

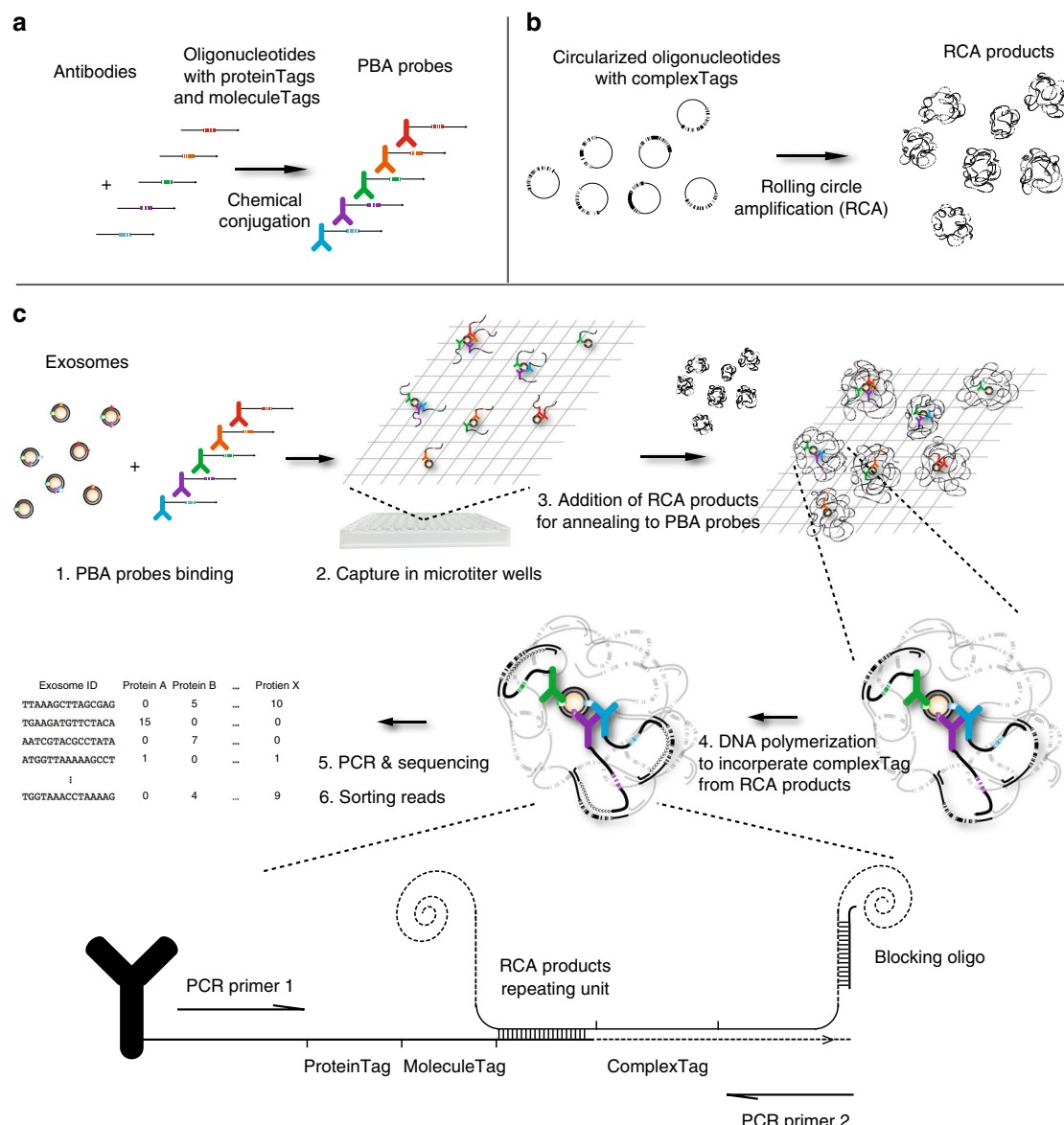

**Fig. 1** Design and workflow of PBA. **a** Preparation of PBA probes by chemical conjugation of antibodies and DNA oligonucleotides containing an 8-nt proteinTag and an 8-nt random sequence moleculeTag. **b** Preparation of RCA products from circularized oligonucleotides comprising a 15-nt random sequence as complexTags, capable of encoding in excess of one billion unique complexTags. **c** To profile surface proteins of exosomes by PBA, exosomes are first incubated with PBA probes, followed by capture of exosomes with bound PBA probes in microtiter wells via immobilized cholera toxin subunit B (CTB). Oligonucleotides on PBA probes brought together by binding the same exosome are next allowed to hybridize to a unique RCA product, followed by enzymatic extension, incorporating the complexTag present in the RCA product along with a standard sequence motif, later used for amplification. To prevent the DNA polymerase from extending across nearby monomers in the RCA products, blocking oligonucleotides are pre-hybridized to the RCA products. Successfully extended DNA molecules on PBA probes are amplified by PCR using the PCR primer1 and PCR primer2 for library preparation, while oligonucleotides on the antibodies or RCA products fail to be amplified by the PCR primer pairs used for amplification. The PCR product were subjected to DNA sequencing to record the numbers of molecules with specific tag combinations, thus revealing the identities of proteins on individual exosomes

proteinTag, while the majority of complexes prepared with mixes of the four oligonucleotides contained more than one proteinTag, with the distribution close to theoretical values. The results indicate that under the appropriate conditions PBA can be used to profile individual protein complexes quantitatively, with minimal risk that proteins from separate complexes are erroneously clustered together (Supplementary note 1, Supplementary Figs. 3–5). Having established that PBA can identify members of complexes in the streptavidin model system, we next investigated the performance of the method for profiling surface proteins on individual exosomes. We prepared four PBA

probes by conjugating CD9 antibodies to two distinct oligonucleotides, forming PBA probes with either proteinTag A (CD9-TagA) or proteinTag B (CD9-TagB), and conjugating CD63 antibody with two other oligonucleotides to obtain either proteinTag C (CD63-TagC) or proteinTag D (CD63-TagD). We then mixed antiCD9-TagA and antiCD63-TagC as probe Set1, and antiCD9-TagB and antiCD63-TagD as the second probe Set2. Figure 3a presents results from an experiment where exosomes isolated from the K562 cell line were incubated separately with the two sets of PBA probes before these were pooled and subjected to PBA with capture in microwells,

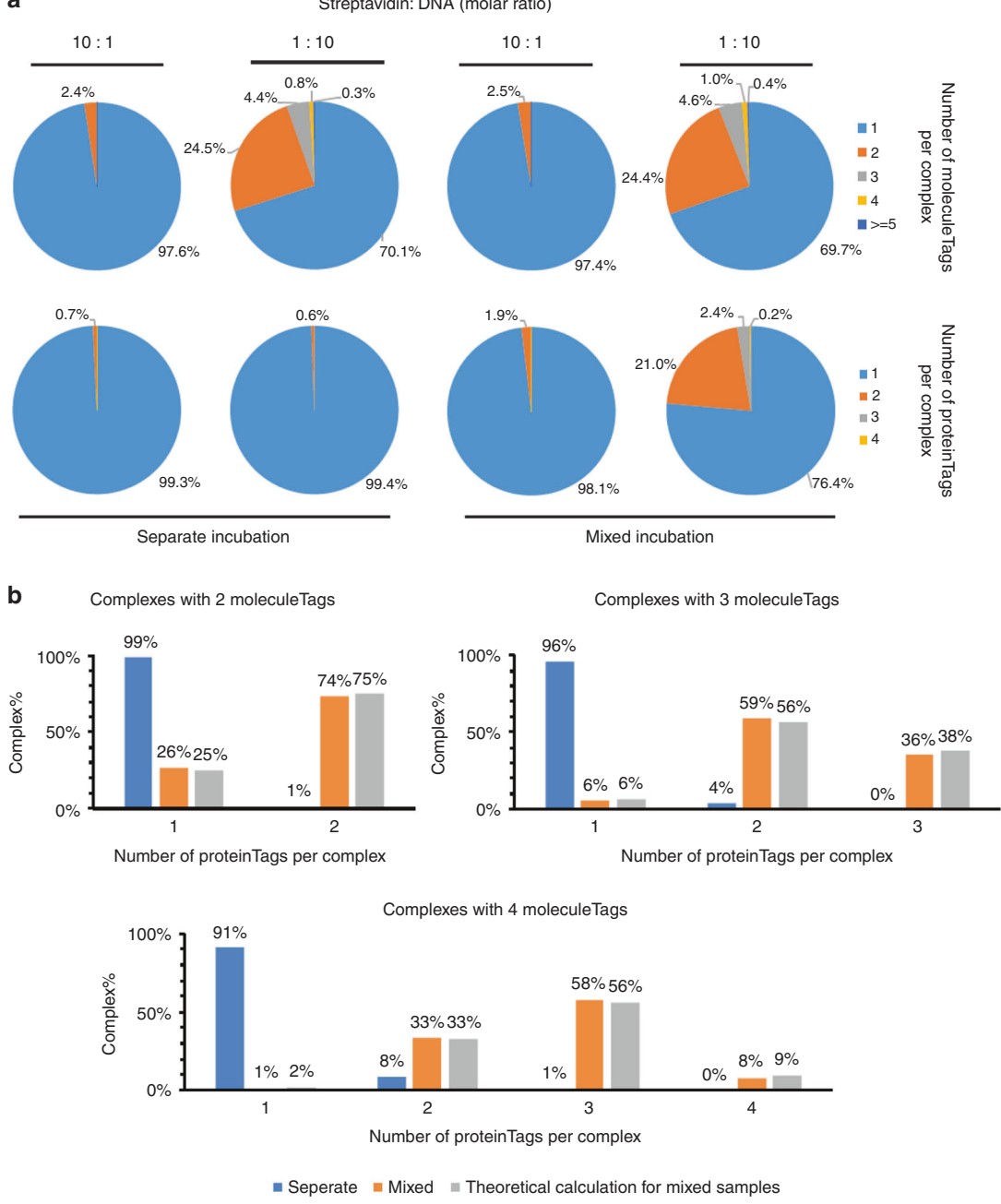

**Fig. 2** Validation of PBA using STV-biotin-oligonucleotide complexes. Tetrameric streptavidin molecules were incubated with four different biotinylated oligonucleotides, either separately or in a mixture of all four oligonucleotides. Streptavidin and biotinylated oligonucleotides were combined at molar ratios of 1:10 or 10:1. **a** The numbers of observed complexes with different numbers of proteinTags and moleculeTags are summarized in pie charts. **b** When the oligonucleotides were in excess, the numbers of complexes from separate or mixed oligonucleotides incubation with 2, 3, or 4 moleculeTags were grouped according to the number of proteinTags per complex. Theoretical ratios were calculated for the mixed sample

annealing to RCA products, extension and sequencing library preparation. In a parallel reaction (Fig. 3b), the exosomes were incubated with all four PBA probes, followed by the PBA workflow. In the reaction with separate probe sets we observed less than 5% of exosomes with PBA probes coming from two different sets (CD9-TagA with CD9-TagB, CD63-TagC with CD63-TagD, CD9-TagA with CD63-TagD, or CD9-TagB with CD63-TagC), while a large majority of the exosomes were tagged with both CD9 and CD63 from the same probe set (CD9-TagA and CD63-TagC or CD9-TagB and CD63-TagD; Fig. 3a). By contrast, in the reaction with mixed probe sets, exosomes were

tagged with probes from both probe sets (Fig. 3b). This again demonstrated the ability of PBA to analyze individual exosomes with minimal risk of false identification, important for characterizing populations of exosomes in biological samples, as a basis to evaluate their distinct diagnostic potential.

**Profiling surface proteins of individual exosomes from different sources by PBA.** Next, we applied PBA to investigate the presence of 38 different human proteins on exosomes from eighteen different human sources—one preparation was isolated

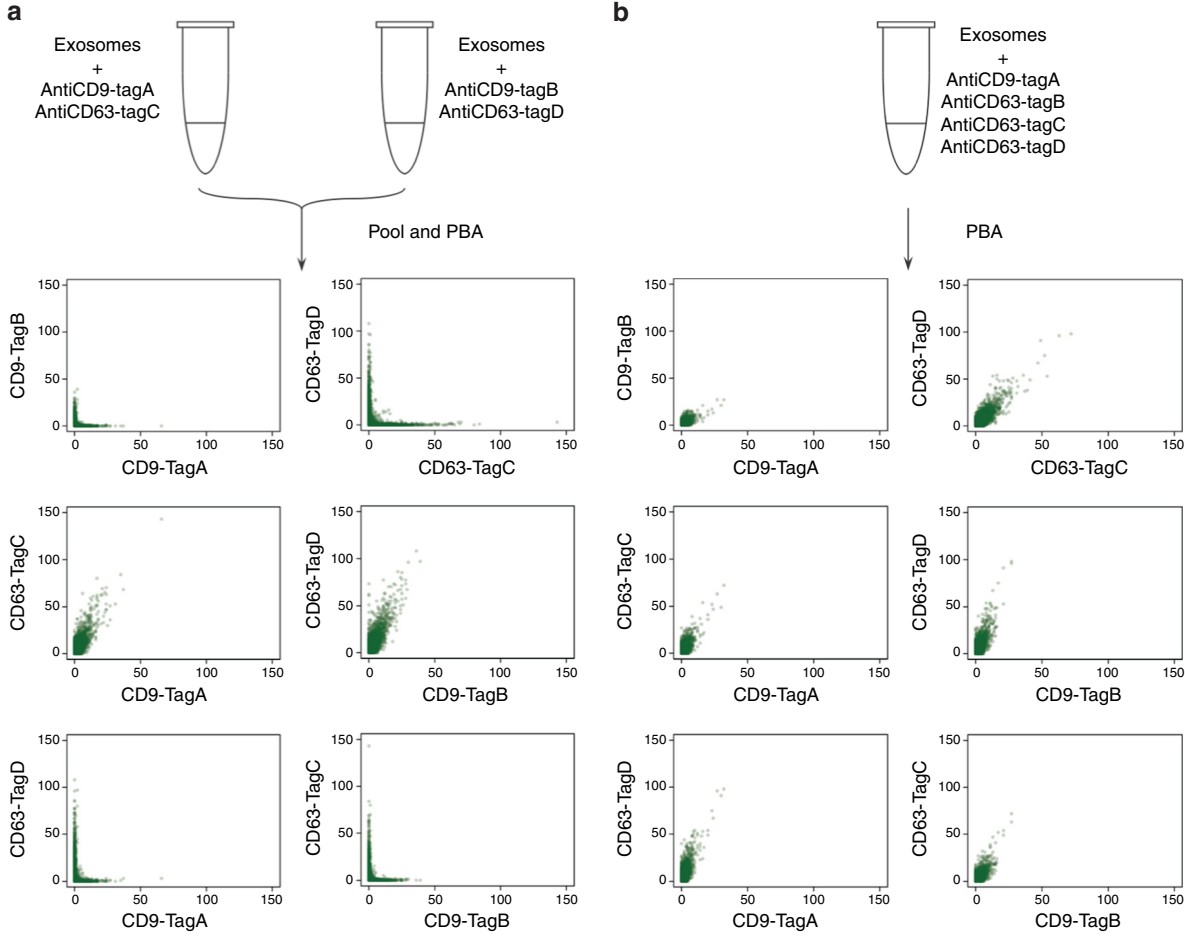

**Fig. 3** Validation of PBA on exosomes using antibodies to CD9 and CD63. **a** Separate incubation: exosomes isolated from the K562 cell line were incubated with two distinct sets of PBA probes separately before they were pooled for PBA. **b** Mixed incubation: exosomes were incubated with four different PBA probes before PBA was performed

from serum from healthy donors (serum exosomes) and one from seminal fluid, referred to as prostasomes, which have been suggested to serve as biomarkers for prostate cancer[12]. Exosomes were also prepared from conditioned media of the 16 human cell lines[10], BLC21, U87MG, A549, HCT116, PC3, HEK293, COLO1, K562, KatoIII, MNK45 MNK7, AGS, MM1, DAUD1, BPH-1, SKNSH, originating from different tissues (see Methods). Each sample was individually exposed to the PBA probes. The antibodies were chosen to include many known exosomal cancer markers, previously used to analyze exosomes from cancer cell lines, with a particular focus on integrin markers, which have been reported to be relevant for cancer metastasis[2].

By counting the total moleculeTags connected with each protein for each sample, the total protein abundance can be recorded (Fig. 4a). Some proteins are present in exosomes from many different samples, e.g., CD151, while some other proteins e.g., CD227 are only expressed in exosomes from a few sources.

We then investigated if exosomes from different sources have distinct surface protein patterns. We performed dimensional reduction of individual exosomes from different sources based on their surface protein compositions using T-distributed Stochastic Neighbor Embedding (t-SNE)[24] (Fig. 4b). For those exosomes from any source where only one protein type was observed in our analysis ($n = 172481$), no good distinction between sources was observed. By selecting exosomes where two protein types were recorded ($n = 17345$), exosome from different sources were better resolved, and some protein pairs were preferentially observed on

exosomes from a single sample source. Lastly, we investigated exosomes for which three or more proteins had been identified ($n = 61882$). Here, exosomes from the same sample sources were seen to colocalize by t-SNE analysis. For some regions, exemplified by dotted circles in Fig. 4b, exosomes from particular sample sources are seen to dominate. This demonstrates that exosomes from different sources are better distinguished when more surface proteins are identified. The PBA analysis does not necessarily detect all proteins present on a given exosomes because of less than quantitative antibody binding and limited sequencing depth.

By comparing the deduced surface protein profiles from individual prostasomes or from exosomes from the K562 cell line with those found in serum, we identified combinations of protein species found mainly in prostasomes (>99%) or in K562 exosomes (>95%) (Fig. 5a). We used this information to mimic analysis of heterogeneous samples, prepared by spiking in different amounts of either purified prostasomes or K562 exosomes or both in a fixed amount of serum exosomes. PBA analysis revealed the expected correlation between proportions of added and recorded exosomes with prostasome- or K562-selective combinations of PBA probes (Fig. 5b–c). Duplicate measurements demonstrated that the measured proportions were reproducible (Supplementary Fig. 6). Pairs of antibodies binding prostasomes as revealed by PBA were also used as probes in solid-phase proximity ligation assays (SP-PLA) providing efficient quantitation (Supplementary Fig. 7).

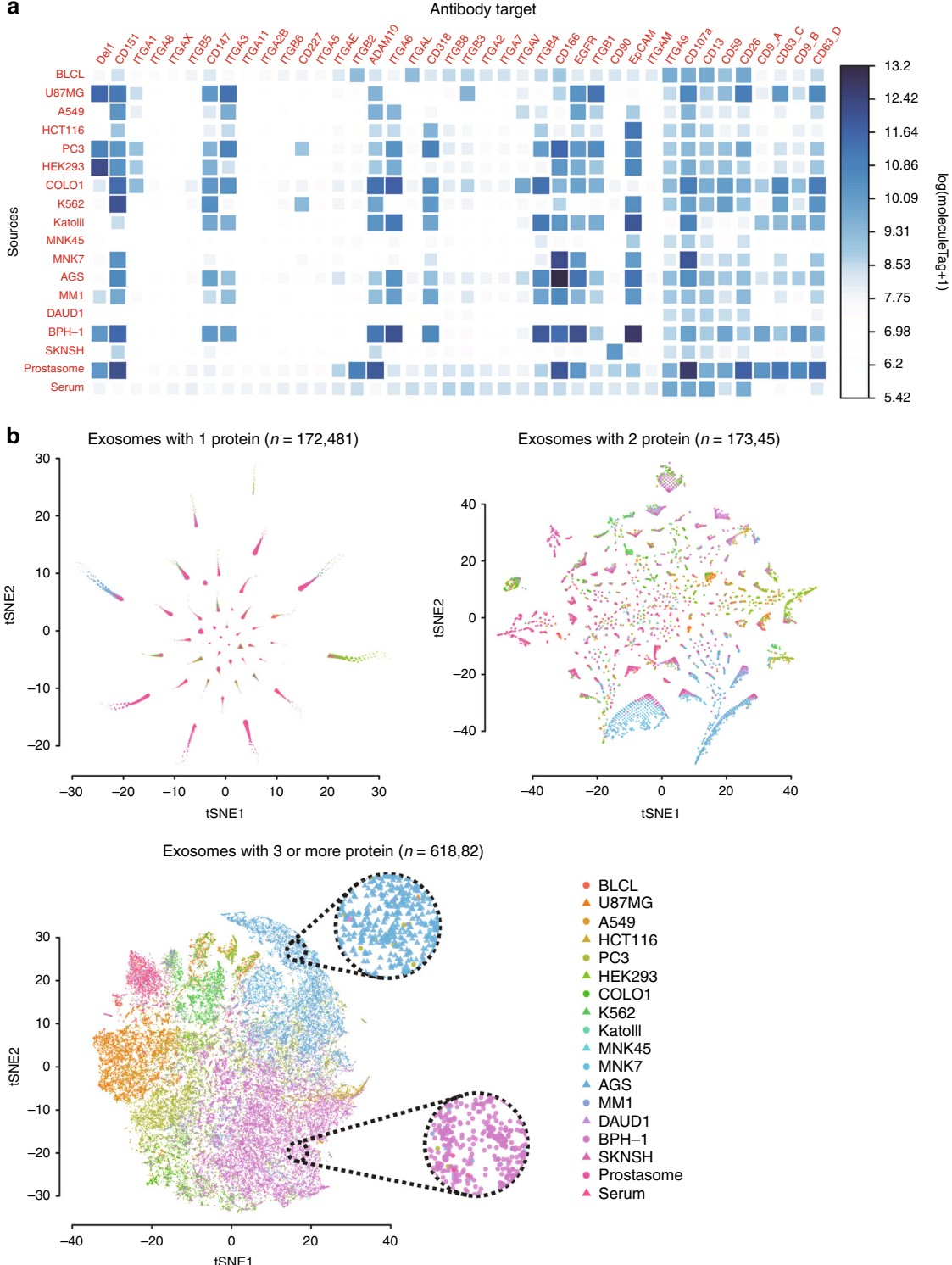

**Fig. 4** Bulk protein counts and visualization of individual exosomes. **a**, **b** Exosomes from 18 different samples were analyzed for the presence of 38 proteins (40 proteinTags) by PBA. **a** Heatmaps representing the total amounts of the proteins by log(moleculeTag + 1) found on exosomes from different sources. **b** Individual exosomes with one identified protein type, two protein types, and three or more protein types were visualized by t-SNE according to their protein compositions, with color and shape of the symbols representing the source of each exosome. Cells from two regions dominated by exosomes from the cell lines AGS and BPH-1 are highlighted in circles with dotted contours

## Discussion

The representation of different classes of exosomes in biological samples may confer valuable diagnostic information, but this information is not readily accessible by current methods for exosome analysis. In biomarker discovery, exosome species that might be of diagnostic value for disease could be present in low abundance among other exosome populations in a sample, obscuring any diagnostic value. Here, we have used DNA sequencing to decode combinations of antibodies with protein-specific DNA tags (proteinTags), binding individual exosomes, by

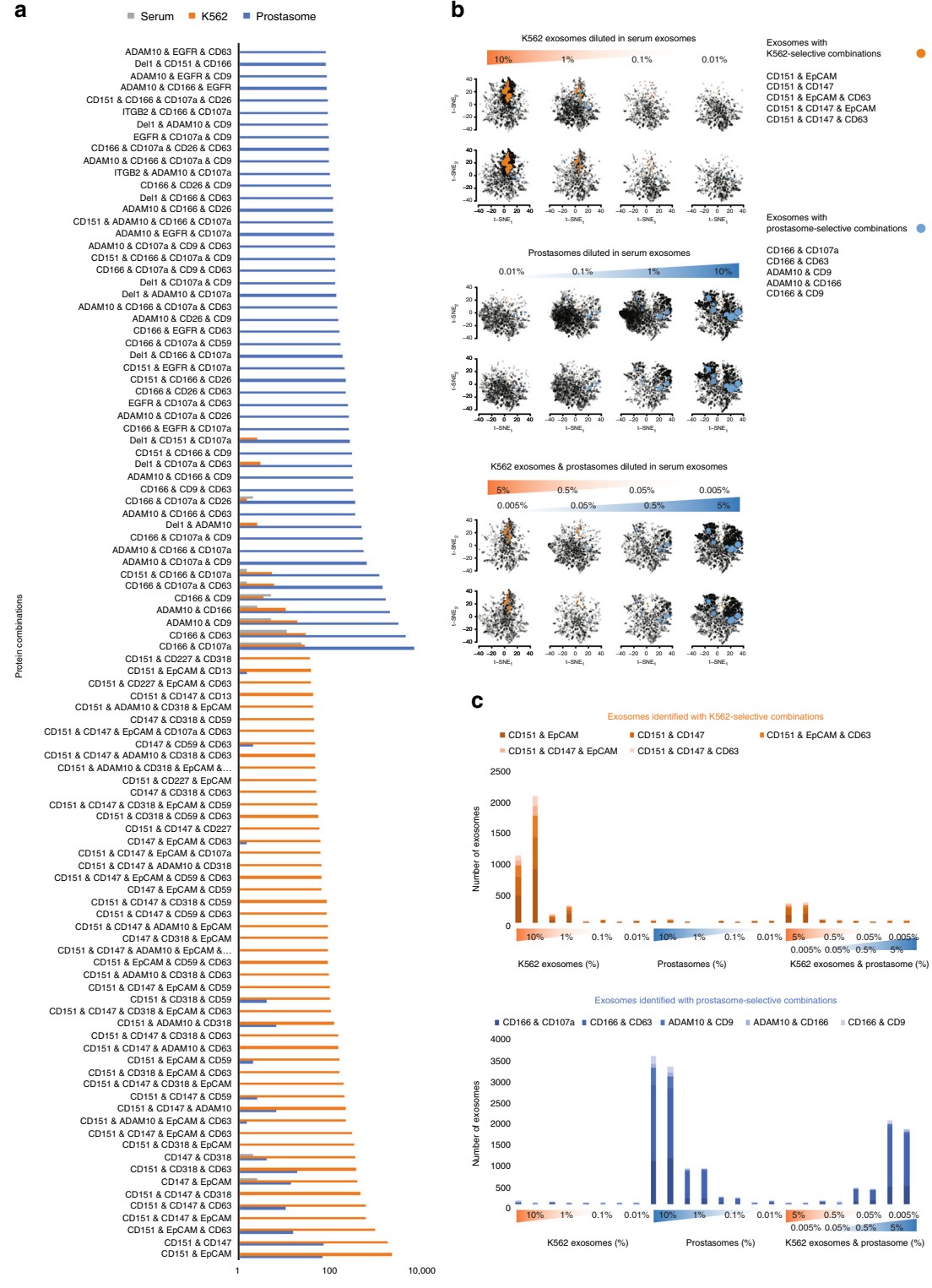

**Fig. 5** Quantification of exosomes from different sources according to their surface protein combinations, as revealed by PBA. **a** Protein combinations selective for either K562 exosomes (orange) or prostasomes (blue), compared to exosomes from serum (gray), were sorted based on the number of exosomes with specific combinations. Here, the top 50 for each exosome are displayed. **b**, **c** Serial dilution of K562 exosomes or prostasomes spiked in serum exosomes were quantified using K562- or prostasomes-selective combinations, respectively. The exosomes identified with either K562- or prostasome-selective combinations are indicated by orange and blue filled circles in the tSNE plots in **b**, and with orange and blue bars, respectively, in the bar plots in **c**. The sizes of the colored circles in (**b**) are proportional to the number of exosomes with the same protein combination. Each two vertically adjacent tSNE plots and two horizontally adjacent bars are illustrating two replicates

associating them via DNA extension with single members of a large repertoire of DNA bundles—RCA products—with multiple copies of a particular random DNA tag—the complexTag. We have used the assays to identify the protein composition of individual exosomes. PBA can identify and quantify large numbers of combinations of proteins from a given set on individual exosomes or any other clusters of proteins.

PBA is performed in microtiter wells without the need for compartmentalization in nanowells[16] or droplets[25–27] or the use of special equipment, by using large numbers of submicrometer-sized DNA bundles generated by RCA, similar in size to the exosomes. Based on the known rate of synthesis by the phi29 DNA polymerase[28], RCA products that represent DNA concatemers with ~200 complements of a 100-nt DNA circle are generated after 10 min of replication. Each of these RCA products thus contains around 200 copies of a unique complexTag that can be combined with proteinTags to identify proteins present on individual exosomes. PBA offers a vast capacity for multiplexing as the current panel of 40 probes, specific for 38 different proteins, can easily be scaled to larger numbers. We demonstrate that PBA can be used to distinguish exosomes through their variable surface protein compositions, enabling analyses of different populations of exosomes in heterogeneous samples.

The limit of detection by PBA depends on the specificity of a given protein combination, the coverage and depth of sequencing, and the affinity and selectivity of antibodies. To increase the probability of detecting elevated levels of exosomes from a given tissue source, targeting tissue-specific proteins and extending the protein panel beyond the current 38 proteins may help to further lower limits of detection. The technique described herein should also be suitable for mapping other higher-order protein complexes, besides those on the surfaces of exosomes or other extracellular vesicles.

In summary, we have developed a technique for analyzing surface proteins of individual exosomes, employing a combination of antibody-DNA conjugates and unique tag sequences repeated in large numbers of unique RCA products. By comparing the profiles of exosomes from different sources, specific surface protein combinations can be found and used for identification and quantification of very large numbers of exosomes that may be released to blood from specific tissues in health and disease.

## Methods

**Assembly of STV-oligonucleotide complexes**. STV was combined with biotinylated oligonucleotides that included proteinTag sequences. Aliquots of recombinant STV (ThermoFisher Scientific) at a concentration of 1 μM were mixed with an equal volume of PBS containing either each one or a combination of all four biotinylated oligonucleotides (Supplementary Table 1) at a total concentration of either 100 nM or 10 μM and incubated at 4 °C overnight. The four separate STV-oligonucleotides complexes were combined into one, and both this pool and the preparation with all four oligonucleotides were diluted to a STV concentration of 1 nM in PBS.

**Preparation of exosomes from cell culture supernatants or from human body fluids**. Gastric cancer cell lines KATOIII (ATCC® HTB-103), AGS (ATCC® CRL-1739), MMK7(RIKEN) were cultured at 37 °C under 5% CO2 in RPMI1640 medium with 10% complete fetal bovine serum (FBS), penicillin-streptomycin, and glutamine. Before exosome isolation, cells were grown in RPMI1640 with 10% exosome-depleted FBS (EXO-FBS, System Biosciences) for 3–4 days up to 95% confluency. Cell culture media were collected and protease inhibitor (Complete Mini, Roche) was added to prevent degradation. Culture media were centrifuged at 3,000 × g for 10 min, and the supernatant was then centrifuged at 10,000 × g for another 10 min at 4 °C to remove cell debris. The supernatant was passed through a 0.45 μm filter and then ultracentrifuged at 100,000 × g for 2 h at 4 °C using a Beckman L8–70M ultracentrifuge. The pellet was resuspended and washed with cold filtered PBS and ultracentrifuged for 2 h again. The final exosome pellet was resuspended in 50–70 μl of PBS containing protease inhibitor and stored at −80 °C until use. The concentration of total protein in isolated exosomes was measured using a Pierce BCA protein assay kit (Thermo Scientific) and a dot-it-spot-it kit

(Maplestone AB, Sweden). Lyophilized exosomes from BLCL (EBV transformed lymphoblastoid B cell line, HBM-BLCL-30/2), K562 (pleural effusion, leukemia chromic myelogenous, HBM-K562–30/2), DAUD1 (human burkitt lymphoma, HBM-DAUDI-30/2), U87MG (human glioblastoma, HBM-U87–30/2), SKNSH (human neuroblastoma, HBM-SK-30/2), HEK293 (human embryonic kidney, HBM-HEK293–30/2), HCT116 (human colon carcinoma, HBM-HCT-30/2), COLO1(human colon carcinoma, HBM-COLO-30/2), A549 (lung carcinoma, HBM-A549–30/2), PC3 (human prostate adenocarcinoma grade IV, HBM-PC3–30/2), BPH-1 (human prostatic hyperplasia), MM1 (human melanoma, HBM-BPH-30/2), and from serum of healthy donors (HBM-PES-100) were all purchased from HansaBioMed Life-sciences LLC.

To purify prostasomes, human seminal plasma was centrifuged at 3,000 × g for 10 min, followed by 10,000 × g for 30 min at 4 °C to pellet cell debris. The supernatant was ultracentrifuged at 100,000 × g for 2 h at 4 °C and the pellet, containing exosomes, was resuspended in PBS. The resuspended pellet was further purified by size-exclusion chromatography on a Superdex 200 gel-filled XK16/70 column (GE Healthcare), followed by density gradient separation (prostasomes were recovered in the density range 1.13–1.19 g/ml). The prostasome concentration was adjusted to 2 mg/ml measured by Pierce BCA protein assay kit and were kept at −70 °C until use.

**Antibodies and oligonucleotides**. The antibody preparation directed against streptavidin was purchased from Thermo Scientific (S10D4). The sources of antibodies directed against surface proteins on exosomes are summarized in Supplementary Table 2. The oligonucleotides used in this study were purchased from IDT or Solulink (Supplementary Table 1).

**Preparation of antibody-oligonucleotide conjugates (PBA probes)**. The conjugation of oligonucleotides to antibodies (Supplementary table 2) was performed as follows: Twenty μg of each antibody was activated by adding 1 μl of 4 mM Sulfo-SMCC (Thermo Scientific) in dimethyl sulfoxide (DMSO; Sigma–Aldrich), and incubating at room temperature (RT) for 2 h. After 1 h of antibody activation, 3 μl of each 5' thiol-modified oligonucleotide at a concentration of 100 μM was reduced by adding 12 μl of 100 mM DTT (Sigma–Aldrich) in 1x PBS with 5 mM EDTA, and incubating at 37 °C for 1 h. The activated antibodies and reduced oligonucleotides were separately purified using Zeba Spin Desalting Plates, 7 K MWCO (Thermo Scientific) according to the manufacturer's recommended procedure. Each purified antibody was then mixed with one type of oligonucleotide (Supplementary Table 1), and directly followed by dialysis in a Slide-A-Lyzer MINI Dialysis Device, 7 K MWCO, 0.1 ml (Thermo Scientific) against 5 l PBS with constant stirring by a magnetic bar at 4 °C overnight. The conjugates were stored at 1 μM antibody concentration in PBS with 0.1% BSA at 4 °C.

**Preparation of RCA products**. Padlock oligonucleotides, including a 15 nt random sequence (100 nM) were ligated into circular DNA strands in the presence of template (100 nM) in 1x phi29 buffer ((Thermo Scientific), 10 mM Mg-acetate, 66 mM K-acetate, 0.1% (v/v) Tween 20, 1 mM DTT) containing 0.01 U/μl T4 ligase (Thermo Scientific), and 1 mM ATP (Thermo Scientific). The ligation was performed at 37 °C for 30 min. To remove unligated padlock oligonucleotides, exonuclease I and exonuclease III (Thermo Scientific) were added to the ligation mix to a concentration of 0.2 U/μl and 2 U/μl, respectively. The reactions were incubated at 37 °C for 30 min and terminated by incubation at 85 °C for 20 min. Then the RCA primer (the same oligonucleotide used as ligation template) was adjusted to a concentration of 100 nM and incubated at 37 °C for 20 min to reanneal to the circular DNA strands. The RCA reactions were initiated by adding d(A, U, G, C)TP at concentrations of 1 mM and phi29 polymerase (Thermo Scientific) at 0.1 U/μl. RCA was performed at 37 °C for 10 min, and terminated by heating at 65 °C for 10 min. The RCA products were kept at −20 °C until use. The RCA products were characterized with Nanoparticle Tracking Analysis NTA (Malvern Nanosight NS300) according to the manufacturer's instructions.

**Capturing STV-biotin oligonucleotide complexes**. For each reaction, 200 ng anti-STV antibodies, diluted in 50 μl 100 mM carbonate buffer (pH 9.6), was added to 8-well RoboStrip (847–0501000103, Analytik Jena) and incubated overnight at 4 °C. After two washes with 100 μl washing buffer (1x PBS with 0.05% Tween20 (Sigma–Aldrich), 50 μl blocking buffer (1x PBS with 1% BSA (Sigma–Aldrich)) was added and the plates were incubated at 37 °C for 1 h. After removing the blocking buffer, pre-assembled STV-oligonucleotides complexes were diluted to 10 pM (according to the concentration of STV) in 20 μl washing buffer, and added to the antibody-immobilized microplates, and incubated for 1 h at RT.

**Probing and capturing exosomes**. For each reaction, 50 ng biotinylated CTB (C-34779, Thermo Scientific) was diluted in 25 μl PBS and incubated overnight in STV-coated PCR tubes (PCR0STF-SA5/100, Biomat). The wells in the plates were then washed twice with 100 μl washing buffer. Exosomes from each sample source or cell line was mixed with all antibody conjugates (20 nM for each), directed against surface proteins, in PBA buffer (PBS, 0.05% Tween 20, 5 mM EDTA, 1 mg/ml salmon sperm DNA (15632011, ThermoFisher Scientific) and 1% BSA) at 4 °C overnight. After incubation, 1 μl of the exosome-antibody conjugate complexes

were diluted in 25 µl PBA buffer, and 20 µl was incubated in wells coated with biotinylated CTB for 15 min at RT. For replicate measurements, 2 µl of exosomes were diluted in 50 µl PBA buffer, and 20 µl were added into two separated wells.

**Proximity barcoding.** The captured STV-oligonucleotide complexes or exosomes with antibody conjugates were washed twice with 100 µl washing buffer. Then 25 µl of the RCA products at a concentration of 1 nM and 500 nM blocking oligonucleotide was added to the reaction wells and incubated at 37 °C for 15 min. The blocking oligonucleotides were used to prevent the DNA polymerase from extending to adjacent copies of monomers in the RCA products containing the complexTags. After two washes with 100 µl washing buffer, 25 µl of T4 DNA polymerase buffer containing 1.25 unit of T4 DNA polymerase and 100 µM d(A, T, G, C)TP was added and the reaction mix was incubated at RT for 15 min.

**Amplification of complex-tagged extension products from the antibody-conjugated oligonucleotides.** The reaction mixes were washed and 30 µl of PCR mix containing 1x Phusion HF Buffer (2 mM MgCl$_2$, 0.2 mM d(A,T,G,C)TP, 1× SYBR Green I, 1% DMSO, 500 nM PCR primers (pba-fwd and pba-rev), 0.02 U/µl Phusion polymerase and 0.02 U/µl Uracil-DNA glycosylase (all from Thermo Scientific) was added to each reaction well. The plate was then transferred to a thermal cycler (MX3005; Stratagene) for qPCR with an initial incubation at 37 °C for 15 min, then 95 °C for 2 min, followed by 30 cycles of 95 °C for 15 s, 60 °C for 30 s, and 72 °C for 30 s.

**Sample indexing and library preparation for sequencing.** After PCR amplification of extension products from PBA reactions, 1 µl of the amplification reaction was spiked in 10 µl of index PCR mix containing 10 µM each of fwd-index and rev-index primer pairs, and the reactions were incubated at 95 °C for 2 min, followed by two cycles of 95 °C for 15 s, 60 °C for 60 s, and 72 °C for 60 s. The indexed PCR products were diluted 20 times into new PCR mixes containing 1x Phusion HF Buffer (Thermo Scientific), 0.2 mM d(A, T, G, C)TP (Thermo Scientific), 1× SYBR Green I (Thermo Scientific), 1% DMSO, 500 nM PCR primers (library-fwd and library-rev) and programmed for an initial 2 min at 95 °C, and 15 cycles of 95 °C for 15 s, 60 °C for 30 s, and 72 °C for 30 s. The PCR products were pooled and purified using QIAquick PCR Purification Kit. The purified DNA was sequenced using MiSeq Reagent Kit v2, 300 cycles by MiSeq or NextSeq Reagent Kit 75SE, NextSeq (Illumina).

**SP-PLA for prostasome detection in 10% human plasma.** Biotinylated CD26 and CD59 were prepared using ChromaLink Biotin Labeling Reagent (Solulink B-1007–105) with 20-fold molar excess of biotin reagent over antibody according to the manufacturer's protocol. Before biotinylation, antibodies were purified using Zeba Spin Desalting Column, 7 K MWCO (Thermo Scientific, 89882) according to the manufacturer's recommended procedure and in PBS at concentration of 2 mg/ml. After biotinylation, the antibodies were purified again with Zeba Spin Desalting Column to remove excess biotin reagents. Dynabeads MyOne Streptavidin T1 (100 µl, 1 mg, ThermoFisher Scientific, 65601) was washed twice with 500 µl washing buffer (PBS with 0.05% Tween 20), and 200 µl of 50 nM CTB-biotin conjugates (C34779) were added to the beads and the suspensions were incubated for 1 h at RT on a rotator. The beads were then washed twice with 500 µl PBST and reconstituted in 200 µl PBS containing 0.1% BSA. PLA probes (CD26-SLC1 and CD59-SLC2) were prepared by mixing 100 nM of biotinylated CD26 and CD59 with 100 nM streptavidin-conjugated oligonucleotides (SLC1 and SLC2, Solulink) at same volume separately and incubated for 30 min at RT. Then two probes were diluted in PLA buffer (1 mM D-biotin, 0.1% purified BSA, 0.05% Tween 20, 100 nM goat IgG, 0.1 µg/µl salmon sperm DNA, 5 mM EDTA in PBS) to 1 nM, incubated for 30 min at RT and mixed at same volume. The final concentration for each probe is 500 pM.

Purified prostasomes were serially diluted in 10% human plasma (in PLA buffer). Each dilution series included a negative control with only 10% human plasma in PLA buffer. Diluted prostasomes (45 µl) were mixed with a suspension of CTB-coated beads (5 µl) and the reactions were incubated for 1.5 h at RT on an end-to-end rotator. The beads were then washed twice in 100 µl PBST using a 96-well magnetic rack. Fifty µl of 500 pM pairs of PLA probes were added into each reaction well, incubated for 1.5 h at RT, and the beads were washed twice in 100 µl PBST. A 50 µl ligation and PCR mixture (1 × PCR buffer, 2.5 mM MgCl$_2$, 0.5 × Sybr Green I, 0.1 µM BioFwd primer, 0.1 µM BioRev primer, 0.1 µM BioSplint, 0.08 mM ATP, 0.2 mM dNTP (with dUTP), 0.03 U/µl AccuStart Taq DNA polymerase, 0.01 U/µl T4 ligase and 0.002 U/µl Uracil-DNA glycosylase) was added to each well. Real-time PCR was performed in MX3005 cycler (Life technologies, CA, USA) programmed at 95 °C for 2 min, followed by 40 cycles of 95 °C for 15 s and 60 °C for 1 min.

**Data analysis.** The BCL files for each sample were converted to fastq formats by using bcl2fastq (Illumina) with pair indexes. Then the complexTag, proteinTag, and moleculeTag were extracted from three fixed segments within each read. Reads with only one count were removed from the analysis. The tags were then sequentially sorted according to complexTag, proteinTag, and moleculeTag using an in-house developed Perl script. The number of exosomes with given combinations of proteins were calculated using an in-house developed R script. The t-

SNE algorithm applied was implemented in the R package Rtsne, and all the parameters were set to default in our pipeline.

**Reporting summary.** Further information on research design is available in the Nature Research Reporting Summary linked to this article.

## Data availability
The data supporting this study are available in Figshare. FASTQ files used in this study are available by https://doi.org/10.6084/m9.figshare.7956023. The protein abundance on individual exosomes of each sample are available by https://doi.org/10.6084/m9.figshare.7963742.

## Code availability
The code used in this study are available upon request.

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

## Acknowledgements
This work was funded by the Swedish Research Council, IngaBritt och Arne Lundbergs Forskningsstiftelse, Knut and Alice Wallenberg Foundation, the European Research Council under the European Union's Seventh Framework Programme (FP7/2007–2013), ERC grant agreement N°. 294409 (ProteinSeq) and Marie Curie ITN grant agreement N° 316929 (GastricGlycoExplorer).

## Author contributions
D.W. conceived the proximity barcoding method. D.W. and J.Y. designed and performed the experiments. Q.S., L.W., L.D. and G.R. prepared the exosomes. Y.C. performed the NTA analysis of exosomes and RCPs. Y.S. constructed the bioinformatics pipeline. D.W., J.Y., X.S., M.T., J.O. and X.Q. analyzed the data and interpreted the results. M.N., U.L., and M.K-M. supervised the project. D.W., J.Y., U.L. and M.K-M. wrote and all authors read and commented on the manuscript.
