## [Peer Review File · Nature Communications]

Reviewers' comments:

Reviewer #1 (Remarks to the Author):

This paper reports the use of proximity ligation assay (PLA) for detection and characterization of single nucleosomes secreted from cells. The authors claim profiling of up to 22 types of nucleosomes from the same sample. While I agree that this is an important topic and the combination of PLA and DNA barcoding/sequencing is a powerful method, I find that the paper underdelivers in several areas that preclude its publication in its current form.

Additional information is needed for evaluating the claim that single exosomes were indeed detected. Specifically, the claim is made that "In PBA, we use micrometer-sized DNA clusters, generated via RCA and similar in size to exosomes, to barcode individual exosomes." The design appears to allow for individual exosomes to produce several different complexTags. This would have the same limitation as nanoplasmonic sensors described in paragraph 2 of the introduction ("The combination of capture and detection antibodies limits the analysis to two protein targets in the same exosome"). Furthermore, no specific consideration is given to the possibility that a pair of probes could bridge over multiple exosomes. In the event that exosomes aren't uniquely barcoded with this procedure, it would be helpful to discuss what benefits this non-unique tagging would offer.

Major comments:

1. The paper is written very poorly. Text and figures are unclear, which made its review particularly difficult. It is difficult to follow their logic in multiple sections of the paper.
2. It is essential to include a detailed description of how oligomers of individual PBA probes combine with circular DNA (and other PBA probes?) to form RCA products. The absence of this description severely limits the relevance of reviewer comments.
3. It is unclear how the authors are able to assert that individual exosomes have been encoded with a specific complexTag. The description lacks sufficient detail to evaluate. As described, it appears an individual exosome may be encoded by several complexTags.
4. The conclusion sentence for figure 2 has been truncated. It is not possible to evaluate the conclusions of this figure.
5. The discussion of figure 4 is too unclear to evaluate the quality of the conclusions. No specific goal is indicated for this figure.
 - a. The discussion indicates a desire to identify exosomes. It is unclear what that would mean in this context. For example, the tissue or origin, or the individual exosomes, ect. In any case, it is difficult to see how exosomes can be conclusively identified without an external validation experiment.
 - b. The discussion of exosome% is unclear and more importantly it does not indicate why such a metric would be useful for the (also unclear) goals of the analysis.
 - c. Figure 4 may benefit from showing less data with a more clear purpose.
6. Care needs to be taken to ensure that individual exosomes on the microtiter plate are not close enough to bridge a single circular oligonucleotide. This can be achieved by using a very low concentration of exosomes.

Minor comments:

1. Figure quality is low, both in regard to design and image quality. Figure 2 has bars that appear above the x-axis. Figure 3 and 4 are illegible. Image resolution is poor, to the point of not being able to evaluate data.
2. sampleTag is undefined
3. Figure 1 has "attaching RCA products." It appears this should say "attaching circular oligonucleotides"
4. Typo in figure 1 "soring" rather than "sorting"

5. Text does not reference figures S1-S3

6. The sampleTag and moleculeTag concepts are already well established in the DNA/RNA sequencing community. Typically they are referred to as "index" and "unique molecular identifier (UMI)" respectively.

Reviewer #2 (Remarks to the Author):

M. Kamali-Moghaddam and colleagues developed a new method to profile surface proteins on individual extracellular vesicles (EVs). After isolation of singles EVs on solid support, they combined three types of barcodes (one allocated to each antibody (protein tag), one allocated to each molecule (molecule tag), and finally one allocated to each antibody complex (complex tag)) to identify several specific combinations of surface markers per donor cell/tissue type. This method enables the identification of specific tissues/cells-derived EVs within a sample of mild heterogeneity. In the future, this method may enable diagnosis thru identification of specific cancer-derived EVs harboring key surface signature.

The method is clever, the controls (in fig 1) are meticulously performed. Reproducibility is addressed in fig S6, and non-specific binding (background) is well addressed. They authors established a proof of concept (detection of CD26-59 positive proteasomes spiked in serum EVs). However, some key points concerning the "limit of detection" of the method in the context of diagnosis need to be addressed.

Major comments:

>The authors demonstrated that prostasomes can be detected when diluted at 1% within serum exosomes (Fig 5b). This "limit of detection" seems high when considering applying the method for diagnosis, when it seems is unlikely that cancer-derived EVs will represent 1% of the totals EVs found in the body fluids. Could the authors at least comment on the limit of detection and what would be required to lower it.

> The authors used a relatively low complexity sample (prostasome spiked in serum exosomes) to validate the method. They should attempt to mix several types of cancers-derived EVs (for instance 3) and then assess if each sub-type (characterized by specific markers combinations) can be detected within a more heterogeneous sample. In addition they should compare and rank the limit of detection for each EV sub-type/ markers combination. Where would the prostasome/CD26-59 combination stand in such a ranking? This will enable the reader to really appreciate how advanced is the method when considering diagnosis applications.

Minor comments:

>Concerning the cell models, the authors should specify if each donor cells originate from human (or other species), and if the selected-antibodies are human specific (or not). If not, antibody/cells inter-species interactions may introduce a strong bias in the analysis and this should be discussed.

>the authors might consider using the term extracellular vesicles instead of exosomes, to avoid debate about the EV-origin.

> the is a typo on Fig 5 d. x axis, should read prostasome (and not proasasome)

Reviewers' comments:

Reviewer #1 (Remarks to the Author):

This paper reports the use of proximity ligation assay (PLA) for detection and characterization of single nucleosomes secreted from cells. The authors claim profiling of up to 22 types of nucleosomes from the same sample. While I agree that this is an important topic and the combination of PLA and DNA barcoding/sequencing is a powerful method, I find that the paper underdelivers in several areas that preclude its publication in its current form.

The PBA technique presented herein is different from our earlier PLA technique, although there are some common elements, namely the use of oligonucleotide-conjugated antibodies, and of rolling circle amplification, here applied to prepare reagents rather than for signal amplification as in the in situ PLA technology. We have now improved Figure 1 and the accompanying text so that the concept of proximity barcoding with rolling circle products (RCPs) can be more easily understood. In the paper we report profiling of up to 38 different proteins on the surfaces of exosomes from a variety of sources. We do not claim to profile any particular number of types of exosomes, since each sample source may contain different types of exosomes, and different samples may well contain similar exosomes.

Additional information is needed for evaluating the claim that single exosomes were indeed detected. Specifically, the claim is made that “In PBA, we use micrometer-sized DNA clusters, generated via RCA and similar in size to exosomes, to barcode individual exosomes.” The design appears to allow for individual exosomes to produce several different complexTags. This would have the same limitation as nanoplasmonic sensors described in paragraph 2 of the introduction (“The combination of capture and detection antibodies limits the analysis to two protein targets in the same exosome”).

We recognize that the question whether we are targeting individual exosomes is very important for evaluating our new technique. In Figure 2 and the new Figure 3, we provide evidence that in the vast majority of cases we are indeed detecting single complexes using the PBA technique. In supplementary notes (Supplementary line 4-48) we provide a theoretical and experimental analysis of conditions to ensure that single complexes are being targeted.

Regarding the comparison with nanoplasmonic sensors this technology is limited to evaluating pairs of proteins per individual exosomes. In the new Figure 4, we show binding by 3 or more antibodies, directed against distinct protein targets, could be detected on individual exosomes using PBA. Accordingly we do not have the same limitation as nanoplasmonic sensors described in the introduction. In our protocol, we capture exosomes via cholera toxin subunit B (CTB) that binds GM1 gangliosides present in the exosome membranes, and hence allows

all exosomes to be captured regardless of their surface protein combinations.

Furthermore, no specific consideration is given to the possibility that a pair of probes could bridge over multiple exosomes. In the event that exosomes aren't uniquely barcoded with this procedure, it would be helpful to discuss what benefits this non-unique tagging would offer.

As discussed in the answer to the reviewer's second comment above, it is important in the PBA technique to operate under conditions where single exosomes meet single RCA products since the aim is to characterize individual exosomes. This consideration is now covered in some detail in the paper in Figure 1-3. An important measure to achieve single exosome resolution is by a process of limiting dilution before the analysis. Only exosomes that come in contact with RCA products can give rise to templates for amplification, while exosomes with antibody-DNA conjugates alone or isolated RCA products fail to give rise to amplifiable products. It is therefore straightforward to achieve conditions where only very rarely two exosomes meet one RCA products or conversely by diluting both components. The detail of this are explained in the Supplementary note, Supplementary figure 3-5.

Major comments:

1. The paper is written very poorly. Text and figures are unclear, which made its review particularly difficult. It is difficult to follow their logic in multiple sections of the paper.

We understand that the details of the paper were difficult to follow. We have modified the figures, including the schematic figure 1, improved the writing throughout, and added experiments illustrated in the new Figures and in the accompanying discussions, to better help the reader understand the procedure and the results that we have obtained.

2. It is essential to include a detailed description of how oligomers of individual PBA probes combine with circular DNA (and other PBA probes?) to form RCA products. The absence of this description severely limits the relevance of reviewer comments.

In the new Figure 1 and in the text (line 99-104) we explain more clearly how the circular oligonucleotides are used to prepare RCA products in advance of the PBA procedure, and we describe how antibody-oligonucleotide conjugates become tagged with sequence information from a nearby RCA product in the assay, allowing the protein composition of individual exosomes to be recorded by DNA sequencing.

3. It is unclear how the authors are able to assert that individual exosomes have been encoded with a specific complexTag. The description lacks sufficient detail to evaluate. As described, it appears an individual exosome may be encoded by several complexTags.

This is a central point of the technique and as described above we now add more

experimental results (the new Figure 3) as well as theoretical modeling (Supplementary note) to explain how individual exosomes can be targeted.

4. The conclusion sentence for figure 2 has been truncated. It is not possible to evaluate the conclusions of this figure.

We are unsure what part of the text the reviewer felt was missing, but it has now been extensively rewritten and we believe it is now easier to understand.

5. The discussion of figure 4 is too unclear to evaluate the quality of the conclusions. No specific goal is indicated for this figure.

a. The discussion indicates a desire to identify exosomes. It is unclear what that would mean in this context. For example, the tissue or origin, or the individual exosomes, ect. In any case, it is difficult to see how exosomes can be conclusively identified without an external validation experiment.

We have replaced the previous figure 4 with a new one, in which we demonstrate that exosomes from different sources can exhibit different surface protein profiles. We trust the amended text is now easier to follow. The aim of our analysis is indeed to investigate the protein composition of individual exosomes from biological samples with many different types of exosomes. In this way it will be possible to distinguish and quantify exosomes that originate from different tissues, perhaps reflecting tissue-specific pathological reactions.

b. The discussion of exosome% is unclear and more importantly it does not indicate why such a metric would be useful for the (also unclear) goals of the analysis.

We removed the measure exosome%. Instead we introduced a tSNE analysis where results of protein measurements for individual exosomes were color-coded according to the sources of the samples. We highlight some clusters of exosomes that predominantly come from a particular sample source.

c. Figure 4 may benefit from showing less data with a more clear purpose. We remove the old Figure 4.

6. Care needs to be taken to ensure that individual exosomes on the microtiter plate are not close enough to bridge a single circular oligonucleotide. This can be achieved by using a very low concentration of exosomes.

It is, indeed, important to use sufficiently low concentrations of both exosomes and RCA products to ensure that the exosomes are uniquely tagged. We addressed this question theoretically in Supplementary Figure 3, and we found that our experimental data (Streptavidin validation experiment) are well in line with the results of modeling. In addition, the new experiment presented in Figure 3 also demonstrates minimal crosstalk between different exosomes. In this new experiment, we separately incubated exosomes with PBA probes

specific for two proteins (anti-CD9-TagA and anti-CD9-TagC), but each tagged with either of two different DNA sequences. The immunocomplexes were then combined and captured in microtiter wells for PBA analysis. Only a small fraction of the complexes included two artificially joined proteinTags. This provides evidence that under our experimental conditions PBA predominantly profiles single exosomes.

Minor comments:

1. Figure quality is low, both in regard to design and image quality. Figure 2 has bars that appear above the x-axis. Figure 3 and 4 are illegible. Image resolution is poor, to the point of not being able to evaluate data.

Images have been reworked, and we combined the figures in the PDF files so that the resolution is now much higher than previously.

2. sampleTag is undefined

We underlined the sequences of the sampleTags in and proteinTag in Table S1..

3. Figure 1 has “attaching RCA products.” It appears this should say “attaching circular oligonucleotides”

The circular DNA molecules containing the 15-nucleotide random sequences not used in the assay but instead they are first used to produce single stranded concatemer RCA products to be used in PBA assays, each containing hundreds of copies of unique complexTags. It is these RCA products that are then added to the immobilized exosomes with their bound antibody-oligonucleotide conjugates. We changed the phrase to “hybridizing RCA products”.

4. Typo in figure 1 “soring” rather than “sorting”

We have changed this in the new Figure 1.

5. Text does not reference figures S1-S3

We have now added references to all the figures in the main and supplementary texts.

6. The sampleTag and moleculeTag concepts are already well established in the DNA/RNA sequencing community. Typically they are referred to as “index” and “unique molecular identifier (UMI)” respectively.

For internal consistency we have kept the nomenclature for the tag sequences that we use in the PBA procedure (moleculeTag, proteinTag, complexTag and sampleTag), but we relate the terms sampleTag and moleculeTag to the corresponding generally used terms index and UMI, respectively (text, line 96-99).

Reviewer #2 (Remarks to the Author):

M. Kamali-Moghaddam and colleagues developed a new method to profile surface proteins on individual extracellular vesicles (EVs). After isolation of singles EVs on solid support, they combined three types of barcodes (one allocated to each antibody (protein tag), one allocated to each molecule (molecule tag), and finally one allocated to each antibody complex (complex tag)) to identify several specific combinations of surface markers per donor cell/tissue type. This method enables the identification of specific tissues/cells-derived EVs within a sample of mild heterogeneity. In the future, this method may enable diagnosis thru identification of specific cancer-derived EVs harboring key surface signature.

The method is clever, the controls (in fig 1) are meticulously performed. Reproducibility is addressed in fig S6, and non-specific binding (background) is well addressed. They authors established a proof of concept (detection of CD26-59 positive proteasomes spiked in serum EVs). However, some key points concerning the "limit of detection" of the method in the context of diagnosis need to be addressed.

Major comments:

>The authors demonstrated that prostasomes can be detected when diluted at 1% within serum exosomes (Fig 5b). This "limit of detection" seems high when considering applying the method for diagnosis, when it seems is unlikely that cancer-derived EVs will represent 1% of the totals EVs found in the body fluids. Could the authors at least comment on the limit of detection and what would be required to lower it.

This is a highly relevant comment as a central aim of our paper is to present a technique to identify rare, tissue-specific exosomes that may serve as disease markers. We address the question both with theoretical and experimental arguments.

As we explain in the discussion (lines 228-233), the "limit of detection (LOD) for a given class of exosomes depends on the availability of combinations of surface proteins that characterize an exosome species. Lower LODs may be possible by expanding the protein panel (in our resubmitted manuscript we have expanded the reagent panel from 24 to 38 antibodies) or including ones directed against protein target of particular interest. We have also increased the sequencing depth to find rare higher-order combination of proteins (more than 4 proteins). To illustrate this we include a new experiment with a slightly different protein panel and a greater number of reads, allowing detection of a little as 0.1% of prostasomes among serum exosomes (Figure 5c).

> The authors used a relatively low complexity sample (prostasome spiked in serum exosomes) to validate the method. They should attempt to miial x several types of cancers-derived EVs (for instance 3) and then assess if each sub-type (characterized by specific markers combinations) can be

detected within a more heterogeneous sample. In addition they should compare and rank the limit of detection for each EV sub-type/ markers combination. Where would the prostasome/CD26-59 combination stand in such a ranking? This will enable the reader to really appreciate how advanced is the method when considering diagnosis applications.

It is known that serum contains a great variety of exosomes, and the serum exosome samples used in this study therefore represent a combination of many classes of exosomes derived from different tissues. To illustrate that also complex mixes can be investigated we diluted both exosomes from the K-562 cell line and prostasomes in serum exosomes in opposite orders of increasing and decreasing concentrations, The results demonstrate the expected increasing or decreasing exosome counts for K-562 or prostasomes as shown in the new Figure 5. In future work we aim to use this method to investigate exosome preparations from body fluids from e.g. cancer patients and controls, but this was beyond the scope of the preset study.

Minor comments:

>Concerning the cell models, the authors should specify if each donor cells originate from human (or other species), and if the selected-antibodies are human specific (or not). If not, antibody/cells inter-species interactions may introduce a strong bias in the analysis and this should be discussed.

The serum sample and cells used in the paper are all of human origin. The antibodies are all directed against human proteins.

>the authors might consider using the term extracellular vesicles instead of exosomes, to avoid debate about the EV-origin.

The materials used in this manuscript are all isolated and purified with standard exosome purification method and thus we have retained the term exosomes.

> there is a typo on Fig 5 d. x axis, should read prostasome (and not proasosome)

We removed the previous Figure 5 and replaced it with a new Figure 5.

REVIEWERS' COMMENTS:

Reviewer #1 (Remarks to the Author):

This work has the potential to be very impactful, particularly regarding exosome research and diagnostics. The revised manuscript has dramatically improved the clarity of experimental design and goals. The primary claims of the paper are: 1) PBA can interrogate individual exosomes, 2) that each exosome can be probed for, at minimum, 38 protein targets, 3) that exosomes can be interrogated from a complex matrix (serum), and 4) these protein targets can identify the tissue of origin of the exosome. Compelling evidence is offered for claims 1), 2), and 3). However, more detail is required to evaluate claim 4). I therefore recommend the authors to submit a revised article that addresses the following issues, which would make this manuscript suitable for publication in Nature Communications:

Major comments:

1. More details need to be provided to evaluate figure 4.

First, there does not appear to be a section in the Methods that describes how the experiment was performed. For instance, were the exosomes pooled and exposed to PBA probes as a group, or were purified exosomes individually exposed to the PBA probes? Since one conclusion is much stronger than the other, it should be made clear to the reader which is the case.

Second, a description of the clustering method should be included. Clustering can be a powerful method for validating data sets such as these, by showing that the clusters that are computationally predicted directly from the data match groups known to the experimenter (in this case, the origin of the exosome). However, tSNE isn't typically used to cluster data. It produces clusters that are sensitive to hyperparameters and this hyperparameter sensitivity could explain why clusters don't appear in the exosomes with 1 and 2 proteins. Therefore, the clustering method requires a description, and an evaluation of how well the clusters match the known groups.

Third, it would help to include a rationale for why the exosomes were split into groups of 1, 2, or 3+ proteins. Is it possible that some exosomes only contain one or two of the proteins targeted by the PBA panel? Considering that most exosomes measure only 1 protein, understanding why could be relevant to assay reliability.

2. Some additional details need to be provided to evaluate figure 5. It shows the identification of protein combinations that can uniquely distinguish K562 exosomes from prostasomes. This point is very clear, however the section where both types of exosome are probed simultaneously doesn't allow the reader to see if the quantification is accurate. For instance, if the measured number of K562 exosomes really is ~ 10 times the number of measured prostasomes when mixed 0.5/0.05% respectively.

3. Some discussion should be included regarding the selection of antibodies for the 38-target PBA panel. Were the antibodies chosen to highlight differences between exosomes? If the rationale is that the panel can identify exosomes, it would help to know which targets matched expectations and which didn't.

4. Several of the figures include features that are unclear.

a. Figure 4a appears to encode the number of moleculeTags by both color and size. As a result of the size-encoding, some squares overlap with each other, preventing the size from clearly encoding anything.

b. Figure 4b does not allow the reader to identify the source of the data, as the colors are very similar, and the overlapping nature of the points prevents distinguishing them by shape. Given the huge number of points used to produce the figure, perhaps a subsampling would be clearer.

c. Figure 5b doesn't have a legend that allows the reader to identify the individual combinations in the signature.

d. In general figure 5 displays a huge amount of information, but without enough context to let the reader identify the most important parts.

5. Some discussion should be included regarding the design of the oligonucleotides. A core aspect of the assay is that "Neither oligonucleotides on exosomes that have failed to encounter an RCA product nor isolated RCA products can give rise to amplifiable products." A figure illustrating how this was accomplished is necessary for the reader to evaluate the suitability of the chosen method.

Minor comments:

1. While the clarity of the manuscript has greatly improved, it could be improved further by explaining why the various analyses/experiments are suitable to prove their intended point. For example, line 160 indicates that the goal of figure 4b is to cluster the data. Line 161 moves directly to explaining what was performed (the data were split into 3 sets) rather than laying out a rationale for why the data need to be split into three to accomplish the clustering. (This pattern also makes it hard to give high-quality reviewer comments).

2. Typo in line 190 – "divers sample"

Reviewer #2 (Remarks to the Author):

In this revised manuscript, the authors have satisfyingly addressed previous comments made by this reviewer, especially the sensibility of the method when analysing more complex samples. The authors also improved the general readability of the manuscript which was pointed out by the other reviewer.

This reviewer still believes that by using the term exosome, the paper will attract unnecessary criticisms that will alter the true message of the study. This method has high impact for extracellular vesicles detection in general, including exosomes, but is not limited to exosomes.

REVIEWERS' COMMENTS:

Reviewer #1 (Remarks to the Author):

This work has the potential to be very impactful, particularly regarding exosome research and diagnostics. The revised manuscript has dramatically improved the clarity of experimental design and goals. The primary claims of the paper are: 1) PBA can interrogate individual exosomes, 2) that each exosome can be probed for, at minimum, 38 protein targets, 3) that exosomes can be interrogated from a complex matrix (serum), and 4) these protein targets can identify the tissue of origin of the exosome. Compelling evidence is offered for claims 1), 2), and 3). However, more detail is required to evaluate claim 4). I therefore recommend the authors to submit a revised article that addresses the following issues, which would make this manuscript suitable for publication in Nature Communications:

Major comments:

1. More details need to be provided to evaluate figure 4.

First, there does not appear to be a section in the Methods that describes how the experiment was performed. For instance, were the exosomes pooled and exposed to PBA probes as a group, or were purified exosomes individually exposed to the PBA probes? Since one conclusion is much stronger than the other, it should be made clear to the reader which is the case.

We are grateful for the several comments by the two reviewers that have allowed us to improve the quality of our paper. We now clarify on line 156 that each source of exosomes was individually exposed to the PBA probes.

Second, a description of the clustering method should be included. Clustering can be a powerful method for validating data sets such as these, by showing that the clusters that are computationally predicted directly from the data match groups known to the experimenter (in this case, the origin of the exosome). However, tSNE isn't typically used to cluster data. It produces clusters that are sensitive to hyperparameters and this hyperparameter sensitivity could explain why clusters don't appear in the exosomes with 1 and 2 proteins. Therefore, the clustering method requires a description, and an evaluation of how well the clusters match the known groups.

We used dimensional reduction rather than clustering, to present the results. We have added sentence about this under "Results" line 164-166, and "Data analysis", line 474-475. The tSNE analyses described in the paper were used to visualize the high-dimensional data in two dimensional space, while clustering was not the purpose of this study. We demonstrate that exosomes from a given source have similar protein profiles in tSNE plots. We also demonstrate that exosomes characterized by specific protein combinations can be identified from tSNE plots for heterogeneous samples (Figure 5).

Third, it would help to include a rationale for why the exosomes were split into groups of 1, 2, or 3+ proteins. Is it possible that some exosomes only contain one or two of the proteins targeted by the PBA panel? Considering that most exosomes measure only 1 protein, understanding why could be relevant to assay reliability.

The PBA analysis cannot be expected to detect all proteins on a given exosome, because of less than quantitative antibody binding and limited sequencing depth. This is now pointed out on line 175-176. The reason we split the exosomes with different number of proteins is to demonstrate that

exosomes for which only a single protein was observed typically fail to be associated with a given sample source, while those with 2, 3 or more proteins are much better resolved according to sample source by providing a more specific profile, as described beginning on line 168-171.

2. Some additional details need to be provided to evaluate figure 5. It shows the identification of protein combinations that can uniquely distinguish K562 exosomes from prostasomes. This point is very clear, however the section where both types of exosome are probed simultaneously doesn't allow the reader to see if the quantification is accurate. For instance, if the measured number of K562 exosomes really is ~10 times the number of measured prostasomes when mixed 0.5/0.05% respectively.

Figure 5 has now been reworked to increase clarity. PBA does not reveal absolute numbers of exosomes, but only proportions among those identified with given sets of antibody conjugates. We have modified the text beginning on line 180 to better explain this.

3. Some discussion should be included regarding the selection of antibodies for the 38-target PBA panel. Were the antibodies chosen to highlight differences between exosomes? If the rationale is that the panel can identify exosomes, it would help to know which targets matched expectations and which didn't.

The antibodies were chosen to include many known exosomal cancer markers, previously used to analyze exosomes from cancer cell lines, with a particular focus on integrin markers, which have been reported to be relevant for cancer metastasis. This important information has now been included on line 156 and onwards,

4. Several of the figures include features that are unclear.

a. Figure 4a appears to encode the number of moleculeTags by both color and size. As a result of the size-encoding, some squares overlap with each other, preventing the size from clearly encoding anything.

It is correct that protein expression was represented by a combination of size and nuance of the squares in Figure 4a to help the reader appreciate differences. We have now adjusted the distance between the squares to avoid the overlaps.

b. Figure 4b does not allow the reader to identify the source of the data, as the colors are very similar, and the overlapping nature of the points prevents distinguishing them by shape. Given the huge number of points used to produce the figure, perhaps a subsampling would be clearer.

The purpose of Figure 4b is to demonstrate the exosomes from the same source tend to have similar protein expression. We do not believe it is possible in this figure to convey which combinations of proteins characterize exosomes from particular sources. This information is instead provided in figure 5.

c. Figure 5b doesn't have a legend that allows the reader to identify the individual combinations in the signature.

d. In general figure 5 displays a huge amount of information, but without enough context to let the reader identify the most important parts.

We have now reworked figure 5 to include fewer antibody combinations so that the reader can more easily identify individual combinations that characterize exosomes from different sources.

5. Some discussion should be included regarding the design of the oligonucleotides. A core aspect of

the assay is that “Neither oligonucleotides on exosomes that have failed to encounter an RCA product nor isolated RCA products can give rise to amplifiable products.” A figure illustrating how this was accomplished is necessary for the reader to evaluate the suitability of the chosen method. Figure 1c has now been modified to more clearly describe the overall design of the DNA sequences in conjugates and rolling circle amplification products, illustrating that the two need to be combined to result in amplifiable products, and we have also modified the legend on line 493-494. We also give the actual oligonucleotide sequences in Table S1.

Minor comments:

1. While the clarity of the manuscript has greatly improved, it could be improved further by explaining why the various analyses/experiments are suitable to prove their intended point. For example, line 160 indicates that the goal of figure 4b is to cluster the data. Line 161 moves directly to explaining what was performed (the data were split into 3 sets) rather than laying out a rationale for why the data need to be split into three to accomplish the clustering. (This pattern also makes it hard to give high-quality reviewer comments).

The text introducing Figure 4b has now been modified to help the reader understand its purpose at line 164-166. As we explain we demonstrate the value of measuring more proteins for each exosomes in order to trace is origin.

2. Typo in line 190 – “divers sample”
We addressed this typo.

Reviewer #2 (Remarks to the Author):

In this revised manuscript, the authors have satisfyingly addressed previous comments made by this reviewer, especially the sensibility of the method when analysing more complex samples. The authors also improved the general readability of the manuscript which was pointed out by the other reviewer.

This reviewer still believes that by using the term exosome, the paper will attract unnecessary criticisms that will alter the true message of the study. This method has high impact for extracellular vesicles detection in general, including exosomes, but is not limited to exosomes. We appreciate the reviewer’s comment that the PBA technique should be applicable to extracellular vesicles generally, and we share this view. However, since the samples used in this study were all treated according to an exosome purification protocol, we have opted to retain the term exosomes in this study.